# Synthetic Inflammation Imaging with PatchGAN Deep Learning Networks

**DOI:** 10.3390/bioengineering10050516

**Published:** 2023-04-25

**Authors:** Aniket A. Tolpadi, Johanna Luitjens, Felix G. Gassert, Xiaojuan Li, Thomas M. Link, Sharmila Majumdar, Valentina Pedoia

**Affiliations:** 1Department of Bioengineering, University of California, Berkeley, CA 94720, USA; 2Department of Radiology and Biomedical Imaging, University of California San Francisco, San Francisco, CA 94158, USA; 3Department of Radiology, Klinikum Großhadern, Ludwig-Maximilians-Universität, 81377 Munich, Germany; 4Department of Radiology, Klinikum Rechts der Isar, School of Medicine, Technical University of Munich, 81675 Munich, Germany; 5Department of Biomedical Imaging, Cleveland Clinic, Cleveland, OH 44106, USA

**Keywords:** image synthesis, inflammatory imaging, deep learning, rheumatoid arthritis, magnetic resonance imaging

## Abstract

**Background**: Gadolinium (Gd)-enhanced Magnetic Resonance Imaging (MRI) is crucial in several applications, including oncology, cardiac imaging, and musculoskeletal inflammatory imaging. One use case is rheumatoid arthritis (RA), a widespread autoimmune condition for which Gd MRI is crucial in imaging synovial joint inflammation, but Gd administration has well-documented safety concerns. As such, algorithms that could synthetically generate post-contrast peripheral joint MR images from non-contrast MR sequences would have immense clinical utility. Moreover, while such algorithms have been investigated for other anatomies, they are largely unexplored for musculoskeletal applications such as RA, and efforts to understand trained models and improve trust in their predictions have been limited in medical imaging. **Methods**: A dataset of 27 RA patients was used to train algorithms that synthetically generated post-Gd IDEAL wrist coronal T_1_-weighted scans from pre-contrast scans. UNets and PatchGANs were trained, leveraging an anomaly-weighted L_1_ loss and global generative adversarial network (GAN) loss for the PatchGAN. Occlusion and uncertainty maps were also generated to understand model performance. **Results**: UNet synthetic post-contrast images exhibited stronger normalized root mean square error (nRMSE) than PatchGAN in full volumes and the wrist, but PatchGAN outperformed UNet in synovial joints (UNet nRMSEs: volume = 6.29 ± 0.88, wrist = 4.36 ± 0.60, synovial = 26.18 ± 7.45; PatchGAN nRMSEs: volume = 6.72 ± 0.81, wrist = 6.07 ± 1.22, synovial = 23.14 ± 7.37; n = 7). Occlusion maps showed that synovial joints made substantial contributions to PatchGAN and UNet predictions, while uncertainty maps showed that PatchGAN predictions were more confident within those joints. **Conclusions**: Both pipelines showed promising performance in synthesizing post-contrast images, but PatchGAN performance was stronger and more confident within synovial joints, where an algorithm like this would have maximal clinical utility. Image synthesis approaches are therefore promising for RA and synthetic inflammatory imaging.

## 1. Introduction

Rheumatoid arthritis (RA) is a widespread autoimmune disorder observed in 0.5–1.0% of the American population, with incidence rates being two to three times higher in women than in men [1]. RA mainly affects the joints, typically the hands and feet, and is characterized by synovial joint inflammation. In the joints it can lead to bone tissue erosions and soft tissue breakdown, often inducing stiffness and debilitating pain, but may also show systemic effects in the skin, heart or lungs if left untreated [2]. It is typically diagnosed through a holistic assessment that begins with a medical history examination, paying particular attention to pain, swelling, peripheral joint pain, and swelling/tenderness, all of which can be indicative of RA. Furthermore, laboratory tests for rheumatoid factor (RF), C-reactive protein (CRP), and erythrocyte sedimentation rate (ESR) are often performed to confirm other RA indications. Lastly, medical imaging plays a crucial role in distinguishing inflammatory phenotypes, providing additional evidence to confirm RA [3]. Once diagnosed, RA is usually treated with Disease-Modifying Anti-Rheumatic Drugs (DMARDs), which see 75–80% of patients attain intended treatment outcomes, but 90% when initiated in the early stages of RA [4]. Robust tools such as imaging are thus necessary for screening and diagnosing RA at early stages, maximizing the odds of successful treatment.

Radiographs have traditionally been the clinical standard imaging modality for RA diagnosis, as their acquisition is quick, inexpensive, and widely accessible, yielding two-dimensional images that are effective in visualizing late-stage bone erosions [5]. In recent years, however, Magnetic Resonance Imaging (MRI) has gained prominence despite its higher costs and longer acquisition time, producing three-dimensional anatomic images with excellent depiction of soft tissues and sharp details [6]. As a result, it has emerged as a superior option for visualizing early-stage bone erosions and bone marrow edema (BME) that can result from RA [7]. An added advantage of MR is the ability to administer contrast agents such as Gadolinium (Gd) prior to scans, altering the magnetic properties of underlying tissue to improve the visualization of numerous pathologies [8]. In RA imaging, a post-contrast Gd MRI can better distinguish active soft tissue RA sites in joints, such as synovitis, from general effusion [9], conveying critical information that conventional MRI cannot provide [10]. However, Gd administration has long-term concerns such as deposition in brain and bone [11,12], is contra-indicated in patient subgroups such as those with renal diseases and pregnant women [13], and, more generally, adds scan time, cost, and patient discomfort to the imaging protocol. As such, if post-contrast MR images could be synthetically generated without Gd administration, the implications for RA diagnosis and other musculoskeletal (MSK) inflammatory conditions or even sarcomas would be significant.

The problem posed by this clinical context is one of “image synthesis”, or the designing of algorithms to generate images from some input. While these inputs can be multimodal, including text or patches of images, the focus here will be on synthesis algorithms that accept full image inputs [14,15]. For image synthesis tasks, deep learning (DL), and particularly convolutional neural networks (CNNs) [16], have taken on an outsized role in recent years. When trained with sufficiently large datasets, CNN filters can be optimized for a given task, with filters in early network layers typically being sensitive to generic features such as edges, while those in later layers are typically sensitive to far more complex, task-specific features [17]. The UNet is a commonly used image synthesis algorithm in which inputted images are encoded by convolutional filters into a low-resolution, high-dimensional representation that is decoded using deconvolutional filters, yielding an output image. Originally designed for segmentation, the UNet has seen substantial application in image synthesis for its ease of training and relatively low dataset size requirements compared to other DL approaches [18]. Another prominent approach is generative adversarial networks (GANs), where an image-to-image translation network such as a UNet (“generator”) is paired with a discriminator network that is trained to distinguish between synthetic and real images [19]. By setting up training as a min-max game in which generator and discriminator networks continually try to fool one another, substantially sharper images can be obtained, although GANs are more difficult to train and are prone to hallucinating artifacts compared to conventional approaches [20]. Other approaches such as variational autoencoders (VAEs) and transformer networks have been investigated in this space [21,22].

These methods have seen considerable application for medical imaging tasks. In brain MRI, image synthesis has been studied for the reduction or elimination of the Gd dosage required for post-contrast tumor imaging. In several studies, standard UNet or encoder-decoder style architectures accepted reduced-dose Gd post-contrast images and/or other MR sequences as inputs, were trained to predict full-dose post-contrast Gd images, and quantified model efficacy through radiologist assessment or the suitability of synthetic images for downstream tasks [23,24,25]. Another approach in eliminating Gd dosage for brain MRI used an innovative training scheme, training a network for tumor detection and passing convolutional feature maps from that network as inputs to a conventional image synthesis architecture. This allowed the image synthesis architecture to focus on pathologic regions when optimizing parameters to produce synthetic post-contrast images [26]. Some approaches beyond image synthesis have also been investigated to eliminate the need for Gd administration. For instance, Gd is administered in cardiac MRI to identify regions of myocardial infarction. Here, DL pipelines have been developed to accept exclusively non-contrast MR images as inputs, localize the left ventricle, extract motion-based features inherent to cardiac MRI, and integrate both to predict if a patient suffered from infarction [27,28]. On the other hand, features from non-contrast MR sequences such as synthetic MRI and diffusion weighted imaging (DWI) have proven effective in differentiating benign and metastatic retropharyngeal lymph nodes, a task that usually requires a post-contrast MRI [29]. Also worthy of mention are recent image synthesis applications in biomedical imaging outside of MRI: in histopathology, standard image synthesis generator networks have been paired with multiple discriminators to generate synthetic stained images, while in microscopy, GAN image synthesis pipelines have been applied for synthetic cell painting, identifying cellular components from brightfield microscopy images [30,31].

These works mark substantial progress, with well-validated frameworks yielding promising results on a wide variety of biomedical image synthesis tasks, including post-contrast MR image synthesis. That said, there are some clear gaps in the literature. For RA imaging, the authors are not aware of any previous work developing post-contrast MR image synthesis algorithms. Such algorithms would have immense clinical utility, synthesizing post-Gd images that could be used to identify synovitis and active inflammation sites in RA patients, while eliminating the risks associated with administering Gd. More generally, Gd is used in brain imaging to identify tumors and distinguish tumor types, while in cardiac imaging it helps identify myocardial infarction sites, among others; in MSK, however, it is administered to image inflammation. Synthetic inflammatory MSK imaging has seen little to no investigation in previous works. Particularly in comparison with brain applications, synthetic Gd dosage reduction in MSK applications, such as wrist imaging, brings about additional challenges such as severe motion artifacts, reduced signal-to-noise ratio (SNR), and considerably smaller datasets [32]. Lastly, despite all these image synthesis works in biomedical applications, efforts to understand the basis of model predictions have been limited; this work would be critical for radiologists to gain confidence in model predictions, a prerequisite for eventual clinical deployment. As such, post-contrast MSK MR image synthesis confers numerous unique challenges that must be managed methodologically, and has been largely unexplored, making it ripe for an initial proof-of-concept study.

This is precisely the niche this work seeks to fill: the purpose of this study was to develop DL pipelines that generate synthetic post-contrast wrist MR images from their pre-contrast counterparts [33], thereby marking the first known effort for synthetic MSK inflammatory imaging. We use image quality metrics to assess the diagnostic and perceptual quality of model-generated synthetic post-contrast images relative to true post-contrast images. We also generate occlusion and uncertainty maps to better understand model performance, making its predictions more trustworthy. More specifically, the contributions and novelty of our work are as follows:To our knowledge, this proof-of-concept study is the first application of DL techniques for generating synthetic post-contrast images for MSK inflammatory imaging.We show that our trained pipelines perform strongly with regards to predicting post-contrast image appearance, particularly in regions afflicted with synovitis, where these models would see the most clinical utility.We investigate the deconvolution operator, checkerboarding artifacts that can be intrinsic to architectures that use it, and how they surface in conventional and adversarial network training schemes.We conduct a rigorous analysis of model predictions, identifying regions in pre-contrast image inputs that were most important to predicted post-contrast images, and regions in which predictions were most uncertain. This provides a straightforward framework that can be used to understand predictions made by image synthesis architectures in biomedical imaging applications.

## 2. Materials and Methods

### 2.1. Study Group

All studies performed in this retrospective study were Health Insurance Portability and Accountability Act (HIPAA) compliant, approved by the UCSF Institutional Review Board (Human Research Protection Program, IRB# 12-10418) and registered under Clinical Trial NCT01773681. Informed consent was obtained from all study participants. Twenty-seven UCSF patients with RA were recruited that met the following criteria: at least 18 years old and fulfilled the 2010 ACR/EULAR criteria for the classification of RA. Patients were treated with either methotrexate or a combination of methotrexate and tumor necrosis factor alpha inhibitors (anti-TNFα) based on RA disease activity; intended sample sizes were thus as large as feasible given the exclusion criteria and the requirements of informed consent from study participants. Data was collected from patients as part of this cohort from 20 March 2014 to 8 February 2018. Patients were imaged at baseline, 3-months, and 1-year follow-up time points, conducting MR imaging, sampling serum to measure ESR, and recording clinical notes at each time point. As the dataset used in this study was from a UCSF clinical trial, data privacy and patient confidentiality concerns prevent its public release, but codes used in generating results can be obtained from the authors upon reasonable request.

### 2.2. MR Acquisition

All patients underwent a standardized protocol that included coronal T_1_ IDEAL scans pre- and post-Gd administration on a 3.0-T wide bore scanner (MR Discovery 750w, GE Healthcare, Waukesha, WI, USA) using 8-channel HD wrist array coils (GE Healthcare, Waukesha, WI, USA). Scans were done with acquisition matrices of 384 × 256 (n = 58) or 256 × 224 (n = 6), a slice thickness of 2 mm, a TR of 457 to 793 ms, and a TE of 10.06–12.48 ms. Complete acquisition parameters for both sequences can be found in Table A1.

### 2.3. Anomaly Segmentations and Evaluations

In post-contrast images, synovitis was segmented in the following synovial joints: intercarpal joints, carpometacarpal joints, the radioulnar joint, and radiolunar joints. Regions with bone marrow edema (BME) were segmented in the following bones: the first to fifth metacarpals, capitate, hamate, lunate, pisiform, scaphoid, trapezium, trapezoid, triquetrum, ulna, and radius. Anomaly segmentations were performed by a radiologist with over 30 years of experience (T.L.) using the Image Processing Package (version 6.43.01) developed by the University of California, San Francisco Musculoskeletal Quantitative Imaging Research Group.

T.L. also quantified synovitis severity for each patient at each time point with the Rheumatoid Arthritis Magnetic Resonance Imaging Score (RAMRIS) for synovitis [34], a 0–9 scale in which a higher score is associated with more severe imaging findings of RA.

Lastly, bounding boxes delineating wrist tissue and background were drawn using the software MD.ai by a radiologist with two years of experience (J.L.), such that reconstruction metrics for synthetic post-Gd images could be evaluated solely in wrist tissue and not be sensitive to textures and noise in background pixels.

### 2.4. Image Preprocessing

Six of 64 acquired imaging volumes had slices that were 256 × 256 pixels, with the remainder being 512 × 512; the slices of these six volumes were upsampled to 512 × 512 using third-order b-spline interpolation. Pre-Gd volumes were then registered to post-Gd volumes with a three-step process: (1) translation, (2) affine, and (3) third order b-spline registration (maximum iterations = 256, 256, 512, respectively; Advanced Mattes Mutual Information [35] criterion for all). B-spline registration was only done for scans where the structural similarity index (SSIM) [36] between pre and post-Gd acquisitions was above 0.5; other scans had motion artifacts so severe that non-rigid registration was not possible. All registrations were performed using SimpleITK 2.0.0 in Python (version 3.7.11) [37,38,39]. Example slices before and after registration can be found in Figure A1. Pixel values in the slices of pre-Gd scans were scaled such that the middle 95% of pixel values were between 0 and 1. The unscaled pixel values in pre-Gd slices that corresponded to 0 and 1 in the scaled slices were also mapped to 0 and 1 in the post-Gd slices, thereby scaling post-Gd slices while preserving the relative enhancement across the volume.

### 2.5. Data Partitioning

The data were partitioned into training, validation, and test datasets, splitting such that all scans from a given patient were in only one of the three datasets. Furthermore, four patients without imaging findings of synovitis were in the dataset (RAMRIS synovitis of 0); splits ensured at least 1 of these patients were in each of training, validation and test. Splits were intended to maintain similar age, BMI, and ESR across the three datasets, but the relatively small overall dataset required some compromise. The full characteristics of the data splits can be found in Table 1.

### 2.6. Network Architecture

All network architectures were implemented in PyTorch (version 1.10.2). Two-dimensional UNet [18] architectures were used as image-to-image synthesizers in our approaches, accepting as input a pre-processed pre-Gd coronal T_1_ IDEAL slice and outputting the corresponding synthetic post-Gd slice. A baseline UNet model was trained, and in a separate pipeline version, an identical UNet was treated as a PatchGAN generator and paired with a PatchGAN discriminator [40]. The PatchGAN discriminator accepted concatenated inputs of the pre-processed pre-Gd slice and either the corresponding synthetic post-Gd slice or the ground truth post-Gd slice, yielding a 16 × 16 output in which each output pixel had a corresponding receptive field “patch” in the concatenated inputs. The 16 × 16 outputs were trained to predict whether synthetic post-Gd generator outputs were real or synthetic. Multiple baseline UNet and PatchGAN generator versions were trained: one set in which all steps of the UNet/generator decoding path used a deconvolution operator, and another in which the deconvolutions were replaced by either a 2 × 2 bilinear upsampling interpolation operator followed by a convolution [41], or just the 2 × 2 bilinear interpolation. The exact network architecture and layers can be seen in Figure 1. Weights for the UNets, UNet generators, and PatchGAN discriminators were initialized randomly to have a mean of 0 and a standard deviation of 0.02.

### 2.7. Training Details

The baseline UNets were trained with a weighted L_1_ loss, as shown below in Equation (1), with loss function variables as follows: n = number of samples; $S_{i}$ = anomaly segmentation mask for slice i; $\hat{y_{i}}$ = synthetic post-Gd image slice; $y_{i}$ = ground truth post-Gd slice. The anomaly segmentation mask $S_{i}$ used to weight the L_1_ loss was calculated as follows: anomaly segmentations were turned into binary masks, any pixel more than 20 pixels from the nearest anomaly was set to a background value $\lambda_{B}$, pixels within anomalies were set to 1, and intermediate pixels were set to a range from $\lambda_{B}$ to 1 based on their Euclidean distance from an anomaly segmentation. A sample distance map can be found in Figure A2.
(1)$$L_{UNet}=\frac{1}{n}\sum_{i=0}^{n}S_{i}(\hat{y_{i}}-y_{i});$$

On the other hand, PatchGAN generators were trained with the same weighted L_1_ loss and a GAN loss, as shown in Equation (2), while PatchGAN discriminators were trained with the loss function shown in Equation (3). Additional variables for these loss functions are as follows: $x_{i}$ = pre-Gd image slice; D(a,b) = PatchGAN discriminator output for concatenated inputs a and b; $\lambda_{L_{1}}$ = anomaly-weighted L_1_ loss weighting for generator; $\lambda_{GAN}$ = discriminator loss weighting for generator. With this loss function setup, the discriminator was trained to predict values of 1 when fed ground truth data and 0 when fed generator predictions, while the generator was trained to do the opposite. For any training batch, the following scheme was followed: (1) synthetic post-Gd generator predictions were calculated; (2) pre-Gd, synthetic post-Gd, and ground truth post-Gd images were used to calculate $L_{Dis}$ and update discriminator parameters; (3) synthetic post-Gd generator predictions and corresponding discriminator outputs were recalculated with new model parameters, $L_{Gen}$ was calculated, and generator parameters were updated; (4) steps (1) and (2) were repeated again to update the discriminator parameters. This approach of two discriminator steps and one generator step per training batch was empirically useful in yielding similar generator and discriminator strength during training.
(2)$$L_{Gen}=\frac{1}{n}\sum_{i=0}^{n}\lambda_{L_{1}}S_{i}(\hat{y_{i}}-y_{i})-\lambda_{\mathrm{GAN}}\log D(x_{i},\hat{y_{i}});$$
(3)$$L_{Dis}=\frac{1}{2n}\sum_{i=0}^{n}\log D\left(x_{i},\hat{y_{i}}\right)-\log D(x_{i},y_{i}).$$

Baseline UNets, PatchGAN generators, and PatchGAN discriminators were all trained with a learning rate of 0.001, an Adam optimizer (β_1_ = 0.5, β_2_ = 0.999), and batch size of 1 to ensure that full batches fit on a single GPU [42]. All pipelines were trained on an NVIDIA Titan Xp 12 GB GPU. For baseline UNet and PatchGAN generator inputs, the following augmentations were done on the training set, each with a probability 0.5: [−2,2] degree random rotation, [−10,10] pixel random translation along both directions in a slice, [−5,5] percent random zoom, and Gaussian noise addition with a mean of 0 and standard deviation of 0.02. Training was done in two stages: initially for 10 epochs in a hyperparameter search to optimize $\lambda_{GAN}$ and $\lambda_{B}$ (more thoroughly described in the following subsection), and finally for 35 epochs with optimized parameters. With 783 pairs of pre and post-Gd slices seen in the training set, this means that 27,405 total slices were seen by all selected models during training (3045 additional slices for validation).

### 2.8. Hyperparameter Search and Model Selection

For each of the four pipelines trained (UNet and PatchGAN, both with and without deconvolutions), grid hyperparameter searches were carried out to optimize the background pixel weighting in segmentation distance maps (0, 0.025, 0.05, 0.075, 0.1, 0.15, 0.2) and $\lambda_{GAN}$ (0.001–0.01, spaced by 0.001). $\lambda_{L_{1}}$ was held constant at 1 for all searches. In hyperparameter searches, models were trained for 10 epochs and model performances were evaluated on the validation set. The most promising parameter set for each of the four pipelines was then trained from scratch for 35 epochs to yield the final models.

The selection of optimal parameter sets was done through a combination of standard reconstruction metrics and visual inspection. For each of the four pipelines, SSIM and normalized root mean square error (nRMSE) were used to screen for top candidate models, whose performance on the validation set was then assessed by visual inspection. The primary criteria for evaluating model performance were (1) the synthesis of new information not obvious from pre-Gd scans, (2) the preservation of sharp textures in synthetic post-Gd scans compared to ground truth post-Gd scans, and (3) the absence of obvious algorithm-generated artifacts that may cause a radiologist to lose confidence in the reconstructed image quality.

### 2.9. Model Performance Evaluation

The assessment of whether to use or omit the deconvolutions in the UNet decoding path was done visually for the UNet and PatchGAN approaches; the best performing models for both methods were then used for a more rigorous analysis. The quantitative assessment of synthetic post-Gd image quality was performed using three standard reconstruction metrics: SSIM, nRMSE, and peak signal-to-noise ratio (PSNR) [43]. Due to the slight misregistration of corresponding slices that may have been present even after previous preprocessing, metrics were presented both with and without slice-wise registration: ((1) 256-iteration translation, (2) 256-iteration affine, and then (3) 512-iteration third order b-spline with a transformation bending penalty of 500, all with the Advanced Mattes Mutual Information criterion). The slice-wise registration was solely for the calculation of model performance metrics; only unregistered model outputs are presented in figures. The reconstruction metrics were evaluated per-volume in the following regions: full imaging volumes, wrist anatomy bounding boxes, and synovial joints. While these metrics do not correlate well with gold-standard radiologist annotations when evaluated on full image volumes or slices, they are widely used in the image reconstruction and image synthesis literature, and thus facilitate easy comparison of model performance with those performing similar tasks [44,45]. Furthermore, our dataset affords us wrist and anomaly bounding boxes; the calculation of these metrics specifically in these regions—one discarding background, and another focusing specifically on tissues of highest clinical interest when administering Gadolinium—can overcome the limitations of these metrics when used conventionally, affording them more clinical significance.

### 2.10. Enhancement Maps

For UNet, PatchGAN, and ground truth post-Gd images, pixels among the top 10% in predicted signal enhancement were identified. Enhancement maps were shown as follows: pre-Gd slice, post-Gd slice, and post-Gd slice with the degree of enhancement overlaid for the most enhancing pixels (top 10%), colored by the predicted extent of the enhancement. For visual consistency, colormap ranges for the enhancement map were calculated with respect to the enhancement observed in ground truth, with the same ranges being used for the maps regardless of algorithmic approach.

### 2.11. Occlusion Maps

For each slice, pre-contrast IDEAL T_1_ images were pre-processed using previously described techniques, which were used as inputs for UNet and PatchGAN generator architectures, generating network outputs. The pixel values were then set to zero in a 32 × 32 occlusion, and the occluded image was fed through the same architecture, recording the absolute difference in predicted pixel magnitude as compared to the unoccluded image. This procedure was repeated for all 32 × 32 occlusions throughout the slice (with a stride length of 8), summing up the predicted changes in pixel magnitudes in an aggregate array and dividing each pixel by the number of occlusions in which it was contained. The aggregate array values were then min-max normalized, divided by pre-contrast IDEAL T_1_ pixel values (to incorporate into resulting maps information for regions other than areas of high pixel intensity), and again min-max normalized, yielding occlusion maps. For display purposes, the maps are thresholded such that only the top 5% of the occlusion map magnitudes were visualized.

### 2.12. Uncertainty Maps

The uncertainty maps of the model predictions were generated by corrupting the latent representations of a given slice [46]. Namely, for 100 iterations, Gaussian noise with a mean of 0 and a standard deviation of 0.5 was added to the encoding path outputs at each of the eight levels (seven layers that were concatenated to the corresponding decoding path levels and the bottom of the encoder). The variance of the predicted pixel intensities from these 100 perturbed latent spaces was then calculated, min-max normalized, and thresholded for display purposes such that only the 15% most variant pixels would display, thereby generating uncertainty maps for each slice.

### 2.13. Statistical Analysis

To assess if synthetic post-Gd scans provided significant improvements over baseline pre-Gd images, 2-sample *t*-tests [47] were conducted. On a per-scanned-volume-basis, these tests compared the metrics of model outputs (nRMSE, SSIM, PSNR) to those of the pre-Gd scanned volumes; a Bonferroni correction [48] was applied when necessary to adjust for multiple comparisons.

## 3. Results

The hyperparameter search results are presented on the validation set, which was used to select optimal values for $\lambda_{GAN}$ and $\lambda_{B}$ in training loss functions. The results from finalized models are presented on the test set, on which finalized models were run just one time. Key demographic information on the test set is available in Table 1.

### 3.1. Model Parameter Selection

The reconstruction performance metrics evaluating the similarity of the synthetic post-Gd model outputs to ground truth were calculated for all 70 tested hyperparameter combinations for each of the four model type configurations (PatchGAN and baseline UNet, with and without decoding path deconvolutions). Sample results are shown for PatchGAN without generator deconvolutions for SSIMs in Table A2, and for nRMSEs in Table A3. Hyperparameter combinations with strong performances in either approach were carried onto a visual inspection of post-Gd synthesis performance, an example of which is shown for several hyperparameter combinations in Figure A3, also for the PatchGAN without deconvolutions. Hyperparameters associated with the selected best models through this process are listed below:PatchGAN, no deconvolutions: $\lambda_{B}$ = 0.05, $\lambda_{GAN}$ = 0.01;PatchGAN, with deconvolutions: $\lambda_{B}$ = 0.15, $\lambda_{GAN}$ = 0.001;UNet, no deconvolutions: $\lambda_{B}$ = 0.05;UNet, with deconvolutions: $\lambda_{B}$ = 0.15.

### 3.2. Utility of Deconvolution Operators in Baseline UNet and PatchGAN Generator Decoders

A comparison of sample synthetic post-Gd slices with and without deconvolutions in the UNet decoding path can be found in Figure 2, while a comparison of synthetic post-Gd slices with and without deconvolutions in the PatchGAN generator decoding path can be found in Figure 3. In baseline UNet pipelines, checkerboarding artifacts were apparent when deconvolutions were used, particularly in regions of relatively homogenous pixel values, such as the muscles around the radius and ulna. When those deconvolutions were replaced by 2 × 2 upsampling and standard convolutions, the checkerboarding artifacts were largely absent. These checkerboarding artifacts were less apparent in both PatchGAN pipelines, but in the version that used deconvolutions, they were evident at the extended boundaries of sharp changes in pixel intensities. Checkerboarding was thus best avoided by PatchGAN and UNet pipelines without deconvolutions, and these pipeline versions were selected as top-performing pipelines for both approaches in the remaining experiments.

### 3.3. Standard Reconstruction Metrics Performance

Standard reconstruction metrics across the test set are shown in Table 2 for full imaging volumes, wrist volumes, and synovial joints. Both synthetic post-Gd volumes had showed significant improvements over pre-Gd volumes in PSNR and nRMSE, with the baseline UNet pipeline also showing significantly higher SSIM. While the UNet baseline model showed stronger performance in all metrics within full volumes and the wrist, the PatchGAN showed stronger reconstruction performance in synovial joints when measured by nRMSE and PSNR.

### 3.4. Comparison of Reconstruction Performance across Synovitis Severity

The image quality metrics for synthetic post-Gd volumes are shown in Table A4 for test set patients without imaging findings of RA synovitis (RAMRIS synovitis = 0, n = 2) and those with imaging findings of RA synovitis (RAMRIS synovitis > 0, n = 5). Though the sample size limits the power of these conclusions, the metrics were slightly stronger for RAMRIS > 0 than for RAMRIS = 0. Visual examples of the reconstructed post-Gd volumes for a RAMRIS = 0 and RAMRIS > 0 patient are shown in Figure 4. In the RAMRIS = 0 patient with no imaging findings of synovitis, the absence of synovial enhancement was captured by both pipelines, whereas in the RAMRIS > 0 patient, UNet and PatchGAN pipelines illuminated similar enhancement patterns in intercarpal regions, with the PatchGAN pipeline depicting sharper enhancement pattern contours, particularly in the muscles and bones.

### 3.5. Enhancement Maps Analysis

Enhancement maps are shown for an example slice for the PatchGAN and UNet models, as well as ground truth, in Figure 5. The enhancement maps show that for the PatchGAN model, general magnitudes of uptake were much more accurately preserved than for the UNet, most notably across intercarpal joints. The predicted enhancement locations were visually very similar for both pipelines.

### 3.6. Occlusion and Uncertainty Maps Analysis

Occlusion maps for the UNet and PatchGAN pipelines in sample test set slices are shown in Figure 6. Encouragingly, occlusion maps for both pipelines show a substantial focus on intercarpal joint regions in terms of their relative importance to the predicted pixel values. Peripherally to the intercarpal joint, the occlusion maps show some focus on muscles as well, perhaps slightly more so for the UNet than for the PatchGAN. On the other hand, the uncertainty maps are shown in an example test set slice for UNet and PatchGAN pipelines in Figure 7. The UNet shows considerable uncertainty in intercarpal joint region predicted pixel values, whereas for the PatchGAN, uncertainty was highest in the background and within the muscles. PatchGAN also showed some uncertainty in predictions within bones such as the radius and ulna, as well as within bone marrow edema regions; notably, however, uncertainty was limited in the synovial joints.

## 4. Discussion

In this work, we developed multiple strong-performing DL pipelines that synthetically generate post-contrast coronal IDEAL T_1_ wrist MR images from pre-contrast coronal IDEAL T_1_ wrist images, marking steps toward synthetic inflammatory imaging of MSK tissues for conditions such as RA. Reconstruction metrics show reasonably strong performances for UNet and PatchGAN pipelines without generator decoding path deconvolutions—PatchGAN nRMSEs in the wrist were 7.68 ± 1.41 (6.07 ± 1.22 after registration, mean ± standard deviation (s.d.)) and for the UNet they were 5.38 ± 0.73 (4.36 ± 0.60 after registration, mean ± s.d.). Standard reconstruction metrics—nRMSE, PSNR, and SSIM—showed the UNet to have superior performance across full volumes and within the wrist, but purely in the synovial joints, where a pipeline like this would see the most utility, the PatchGAN outperformed the UNet. These findings provide yet additional evidence to a growing body of literature which suggests that standard reconstruction metrics do not provide great correlation with clinically useful metrics when evaluated in a classical fashion (across an entire tissue) [44,45,49]. This, in addition to a perceptually stronger performance replicating sharper textures (particularly within muscles and bones, but at times in the synovial joints as well), shows the PatchGAN pipeline without deconvolutions to be the strongest tested version and with the most potential for eventual clinical use with further development. Additionally, enhancement maps showed that while both pipelines exhibited similar performance in identifying the location of the top 10% of enhancing pixels, the PatchGAN did a substantially better job in preserving the enhancement magnitudes. These trends particularly held in the muscles and vessels, but also in many synovial joints.

To build clinicians’ trust in medical image processing algorithms, experiments such as the proposed occlusion map and uncertainty analyses are vital to address the criticism of deep learning algorithms being “black boxes”. These techniques yielded notable insights in the PatchGAN and UNet pipelines: occlusion maps showed that both pipelines focused heavily on intercarpal regions and synovial joints as a basis for generating model predictions. At the same time, uncertainty maps yielded diverging conclusions: whereas the PatchGAN was most uncertain in background, muscles, and within bones, the UNet pipeline was the most uncertain within the intercarpal joints themselves. Given that intercarpal joints—and more generally synovial joints—are where a synthetic inflammatory imaging algorithm would see maximal utility in RA imaging, it is extremely encouraging that the PatchGAN based much of its predictions on the intercarpal joints and was relatively confident in its predictions. This, combined with the superior reconstruction metrics obtained in synovial joints by the PatchGAN as compared to the UNet, confirms it to be the pipeline with the most potential for clinical utility, and indicates that the combination of a GAN and a focused, ROI-based loss can yield promising results for optimizing image synthesis algorithms. Uncertainty and occlusion map approaches such as those applied in this work are straightforward to implement and can be extended to other deep learning applications such as image synthesis, image segmentation, and image reconstruction. In doing so, they can make the findings of such algorithms easier to interpret while providing valuable insights into how they work. From a clinical perspective, they can not only build trust in algorithm outputs, but also direct a radiologist’s attention to uncertain regions in an image that require closer examination.

The exploration of architectural designs also yielded interesting insights. Checkerboarding artifacts have long been reported as a shortcoming of CNNs, and more specifically UNets, with many strategies being proposed to mitigate them [50,51,52]. Our investigation of UNet pipelines with and without one such mitigating strategy—replacing deconvolutions with interpolation and standard convolutions—showed checkerboarding artifacts to be widespread in larger areas of relatively homogenous pixel intensity with the standard deconvolutions, but absent with the mitigating strategy implemented. When paired with a PatchGAN discriminator, even a UNet generator with deconvolutions resolved the checkerboarding artifacts in larger homogenous pixel intensity areas, but saw minor checkerboarding emerge at the boundaries between pixel intensities. Checkerboarding artifacts are thus intrinsic to the standard UNet architecture, and among the tasks a discriminator must learn in adversarial training is their removal. When deconvolutions are replaced with interpolation and standard convolutions, the artifact removal responsibility is simplified for a GAN discriminator, in theory allowing the discriminator to focus on more minute differences between real and synthetic images and, thus, possibly producing stronger synthetic images. These lessons can be translated to GAN training strategies in other settings—training schemes may yield stronger results after the thorough inspection of generator architectures to ensure that obvious artifacts are not intrinsic to the network design.

It is clear from our work that larger sample sizes are needed to derive statistical conclusions with more power and to assess algorithm efficacy stratifying by race, RA status, and others. However, this study nonetheless serves as a strong proof-of-concept indicating the potential for DL algorithms to synthesize post-contrast images for inflammatory imaging in MSK applications. Importantly, these algorithms can synthesize images in a negligible amount of time, essentially providing free information for radiologists examining inflammation, even for the many patients for whom contrast MR sequences would otherwise not be prescribed. With additional validation, and through building clinicians’ trust in these algorithms, they can allow for safer, more comfortable, and less time-consuming RA diagnosis and treatment through synthetic imaging. Beyond the proof-of-concept wrist RA post-contrast synthesis, this work can seed new efforts in other MSK applications such as synthetic RA imaging in other joints [53], synthetic screening for sarcoma [54], more thorough investigations associating contrast and non-contrast MRI of Hoffa’s fat pad with pain [55], larger cohort studies assessing bone perfusion [56], and safer imaging techniques to diagnose spondylodiscitis [57]. In all these applications, Gd is administered in standard imaging protocols, so similar datasets can be curated and used to train synthetic post-contrast imaging algorithms to reduce and hopefully eliminate the need for Gd administration. Furthermore, validated algorithms could synthesize post-contrast images from existing large datasets such as the Osteoarthritis Initiative (OAI), K2S, and fastMRI+ to allow for large cohort studies to facilitate a better understanding of inflammation [58,59,60].

This study had several limitations. Ideally, there would be a true comparison of algorithm performance in patients with and without RA to ensure strong performance in both, but ethical considerations prevented us from administering Gd to healthy controls. In the absence of this, we used RAMRIS scores to stratify RA patients into subgroups of those with and without imaging findings of RA for a pseudo-control study, but this is not a true control study. Furthermore, the desire to compare algorithm performance in patients with and without imaging findings of RA in a pseudo-control study, combined with the small dataset size, led to some imbalance in demographic characteristics across training, validation, and test datasets. Namely, test set patients had the least severe RA. Additionally, pre-Gd coronal IDEAL images were registered to corresponding post-Gd images in data preprocessing. Radiologist anomaly segmentations were performed only on post-Gd images, so doing so allowed segmentations to be used in weighting loss functions and assessing model performance in anomalous regions, but this registration step would not be possible at the inference time. There was thus a tradeoff between optimizing trained algorithms for strong performance in synovial joints and using a realistic workflow for eventual clinical utility; the authors viewed the former as more important in a proof-of-concept approach. Lastly, standard imaging protocols would typically use T_1_ pre-contrast scans and fat-saturated post-contrast T_1_ scans for RA imaging. Our approach used IDEAL scans before and after contrast administration, as these sequences were available in our dataset, but for true clinical translation an algorithm should be trained on these other sequences. The structure of our dataset thus conferred many limitations on our work, but nonetheless, it represents a meaningful first step towards making synthetic inflammatory imaging a larger research focus for the MSK community.

## 5. Conclusions

To the best of the authors’ knowledge, our work marks the first concerted effort at leveraging DL for synthetic inflammation imaging for an MSK application. We developed PatchGAN and baseline UNet pipelines that showed strong performance synthesizing post-contrast IDEAL T_1_ images from corresponding pre-contrast IDEAL T_1_ images, with the PatchGAN pipeline outperforming the UNet in synovial joints, generating more accurate and confident predictions where a model would have the most utility. The PatchGAN also showed magnitudes of signal enhancement that more closely match that of ground truth images and retained sharp textures in synthetic images. As such, the PatchGAN model was particularly promising in synthesizing post-contrast inflammatory images, and with further development, it could reduce or eliminate the need for Gadolinium administration in treating patients with RA. There are numerous future directions for research: (1) more sophisticated GANs such as CycleGAN can be implemented to improve the sharpness in reconstructed images; (2) generator architectures that learn registration transforms and predict images can also be investigated, eliminating the need for the registering of pre-Gd images to post-Gd images, which would not be possible at inference time in the clinic; (3) investigating other loss functions, such as other types of GAN distances; and (4) assessing model robustness by inferring from conventional wrist coronal T_1_ scans to evaluate predicted post-contrast scans on conventionally used clinical sequences in inflammatory imaging. For substantial progress, however, the MSK field will require concerted efforts to curate larger datasets for inflammatory RA conditions that will allow for more statistically powerful conclusions, more complicated models, and comparisons across population subgroups. Our hope is that the promise of our results can motivate efforts to do so.

## Figures and Tables

**Figure 1 bioengineering-10-00516-f001:**
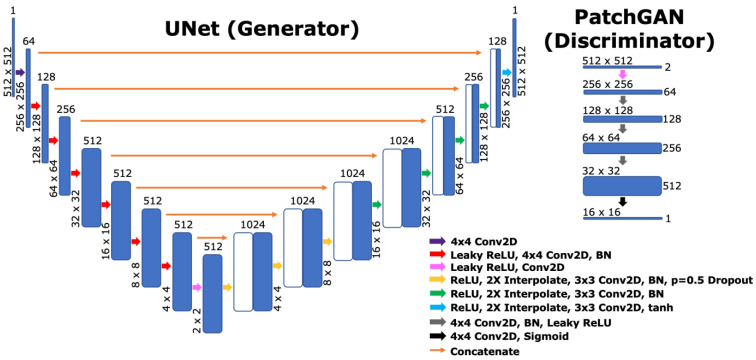
Network Architectures. The baseline UNet and PatchGAN generators used identical architectures, while the PatchGAN pipeline also trained a discriminator whose architecture is pictured. All generator encoding path convolutions had a stride of 2 and a padding 1, while all decoding path convolutions had a stride and padding of 1. The first three discriminator convolutions had a stride of 2 and a padding of 1, while the final two had a stride and a padding of 1. For PatchGAN and UNet pipelines with deconvolutions, all “2X interpolate, 4 × 4 Conv2D” steps would be replaced by 4 × 4 2D transposed convolutions with a stride of 2 and a padding of 1. All leaky ReLU layers had a negative slope of 0.2.

**Figure 2 bioengineering-10-00516-f002:**
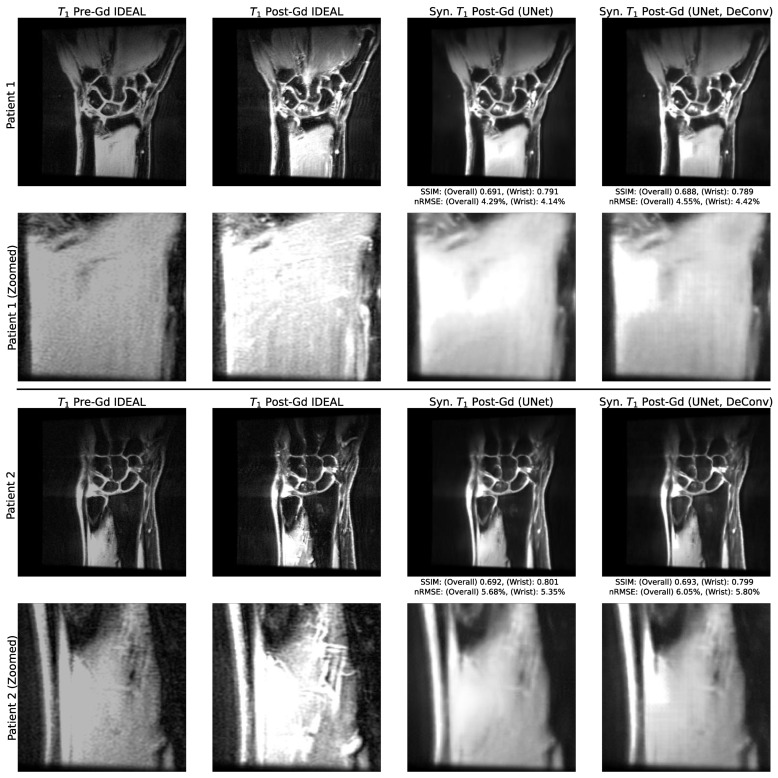
Network Performance with and without Deconvolutions in Decoding Path of Baseline UNet. The performance on example test set slices for baseline UNet, with and without decoding path deconvolutions, with zoomed insets. The use of decoding path deconvolutions in baseline UNets induces checkerboarding artifacts in larger regions of relatively homogenous pixel values, such as the forearm muscle insets (particularly evident in patient 1). When replaced with convolution and interpolation operators, these artifacts were substantially mitigated, making this the preferred architecture when training baseline UNets.

**Figure 3 bioengineering-10-00516-f003:**
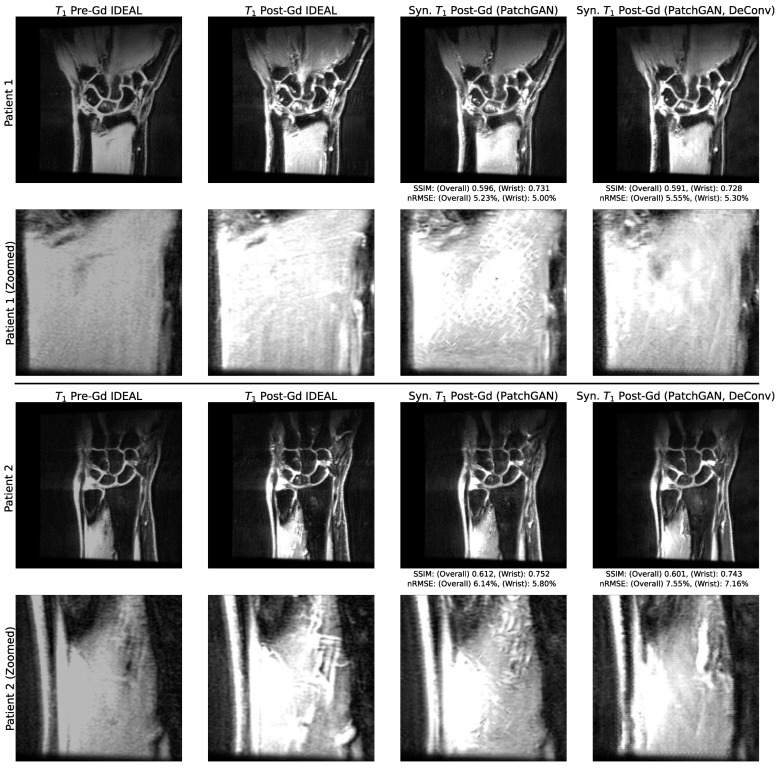
Network Performance with and without Deconvolutions in Decoding Path of PatchGAN Generator. The performance on example test set slices for PatchGAN pipelines, with and without generator decoding deconvolutions, with zoomed insets. At sharp transitions in pixel intensities, such as intersections of the radius and ulna with muscles displayed in insets, clear checkerboarding is observed when deconvolutions are used. This was substantially reduced when deconvolutions were replaced with convolutions and interpolation; PatchGAN generators with these decoding path operations were thus used when training PatchGAN pipelines in the remainder of this paper.

**Figure 4 bioengineering-10-00516-f004:**
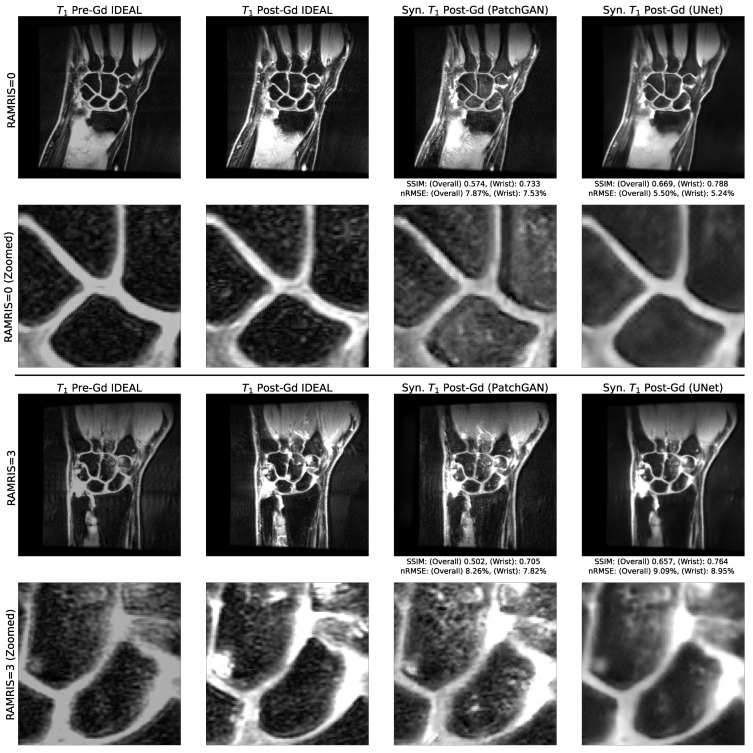
Visual Comparison of Reconstructed Post-Gadolinium Images with and without Imaging Findings of RA. Two example test set slices reconstructed by baseline UNet and PatchGAN pipelines for patients with and without imaging findings of RA (RAMRIS = 3, RAMRIS = 0, respectively). There was little to no enhancement in the synovial joints of the RAMRIS = 0 patient, which is captured by both pipelines, as seen in the zoomed insets. In the RAMRIS = 3 patient, the contours of enhancement in the zoomed inset were captured well within the intercarpal joint for both pipelines, with noise distribution patterns better reconstructed by the PatchGAN. The reconstruction performance thus shows promise for patients with and without imaging findings of RA.

**Figure 5 bioengineering-10-00516-f005:**
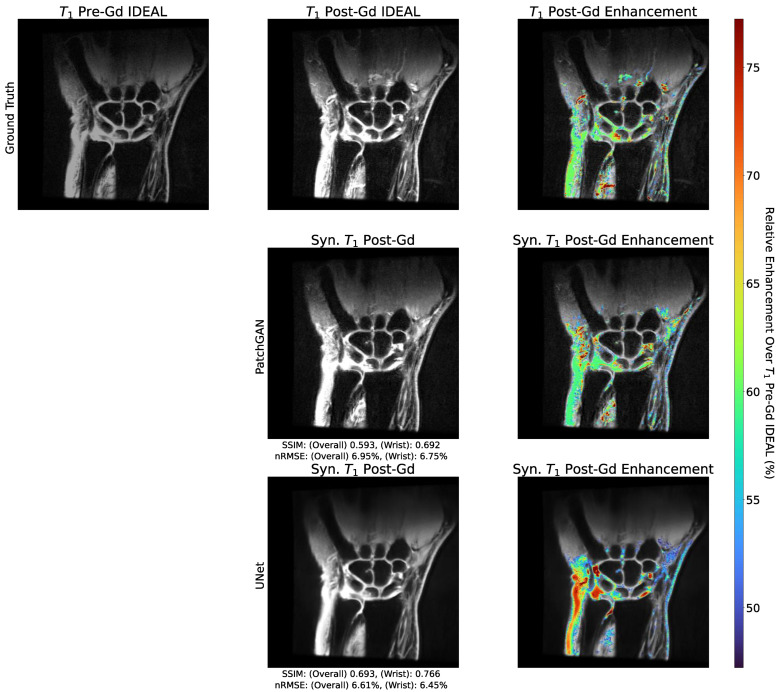
Predicted Gadolinium Enhancement Maps with PatchGAN, UNet, and Ground Truth Models. Enhancement maps were generated by identifying the magnitude of pixel intensity increase from synthetic or ground truth Post-Gd slices compared to corresponding pre-Gd slices, and by highlighting the top 10%. While the performance in preserving the location of these top 10% of enhancing pixels was similar for the baseline UNet and PatchGAN, the enhancement magnitudes were far better preserved globally by the PatchGAN, including intercarpal regions susceptible to synovitis. These maps reflect the long-term vision of a pipeline like this: given a pre-Gd scan, the algorithm can identify locations susceptible to synovitis and distinguish active inflammatory sites from general effusion with additional model development.

**Figure 6 bioengineering-10-00516-f006:**
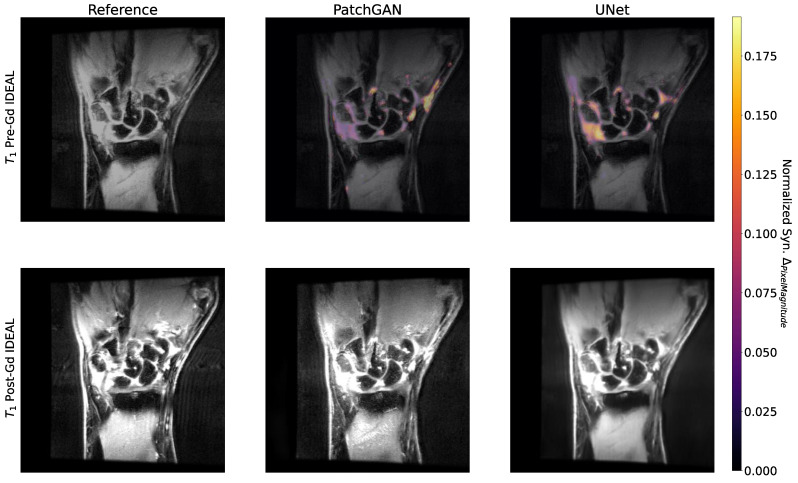
Occlusion Maps for PatchGAN and UNet Pipelines. Occlusion maps were generated for PatchGAN and UNet by occluding 32 × 32 patches of the input slices and assessing changes in predicted pixel values compared to unoccluded slices. Occlusion maps were then normalized by pre-Gd pixel intensities and thresholded to identify hotspots most impactful in model predictions. For UNet and PatchGAN, hotspots primarily included intercarpal joint regions. Particularly for the UNet, the maps also showed some emphasis on the forearm muscles. Given that the synovial joints are where an inflammatory imaging algorithm would see the most utility, the fact that both algorithms placed heavy emphasis on the intercarpal regions was promising, indicating that both focused on synovitis-relevant regions to make predictions.

**Figure 7 bioengineering-10-00516-f007:**
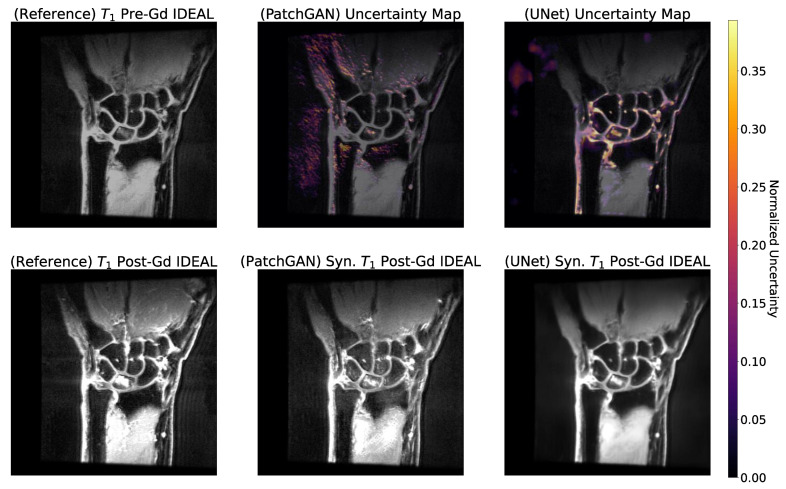
Uncertainty Maps for PatchGAN and UNet Pipelines. Uncertainty maps for the PatchGAN generator and baseline UNet were generated by corrupting the latent space of all encoding path outputs, adding Gaussian noise with a mean of 0 and a standard deviation of 0.5 for 100 iterations, and calculating the variance in the predicted pixel magnitudes for each output pixel across these iterations. The most variant pixels were designated as the most uncertain ones. For the PatchGAN, the uncertain regions were mainly in the background, muscles, and within bones. For baseline UNet, the uncertainty maps placed a heavy emphasis on the intercarpal joint, with some residual highlighting of background. In conjunction with occlusion maps, PatchGAN generator predictions were more confident and less uncertain within intercarpal joint regions compared to the baseline UNet. Considering that the intercarpal joint is crucial for synovitis diagnosis and is where both algorithms would be the most useful, the PatchGAN’s confident predictions within it were promising.

**Table 1 bioengineering-10-00516-t001:** Full Dataset and Splits Information. Demographics and patient information for the entire dataset and splits into training, validation and test. All data are presented as mean ± 1 s.d. The dataset consisted of 27 patients diagnosed with RA, each of whom were scanned up to three times (baseline, 3-month, and 1-year follow-up after one of two treatments). Data splitting was done at a patient level while ensuring each of the training, validation and test datasets included at least one patient with a RAMRIS synovitis of 0. The small dataset size and splitting conditions caused slight imbalances in demographic and health variables across the splits.

	Train	Validation	Test	Full
Age	53.38 ± 13.50	45.94 ± 16.16	52.12 ± 18.60	52.41 ± 14.65
BMI	29.35 ± 8.90	25.32 ± 3.06	28.33 ± 1.26	28.79 ± 8.03
ESR [mm/h]	29.06 ± 26.07	32.00 ± 24.00	27.00 ± 20.12	29.05 ± 25.32
RAMRIS Synovitis	4.57 ± 2.13	2.33 ± 2.62	1.67 ± 1.25	4.00 ± 2.37
Slices	783	87	105	975
Volumes	51	6	7	64

**Table 2 bioengineering-10-00516-t002:** Coronal IDEAL Post-Gd T_1_ Image Synthesis Performance for Select Pipelines. Standard reconstruction metrics of the PatchGAN and baseline UNet pipelines were evaluated on a per-patient basis within the test set (n = 7) for entire imaging volumes (“full”), wrist tissue in each volume (“wrist”), and synovial joints. Metrics were calculated with and without three-stage nonlinear registration of synthetic post-Gd volumes to ground truth. UNet pipelines reflect the stronger bulk reconstruction metrics in full volumes and within wrist tissue, whereas the PatchGAN pipeline shows stronger performance in the synovial joints in which an algorithm like this would have most clinical utility, and was therefore the stronger model. The 2-sample t-tests with Bonferroni correction showed that nearly all pipelines offered significantly better reconstruction metrics than Pre-Gd baselines (n = 7; * *p* < 0.05, ** *p* < 0.01, *** *p* < 0.001).

		Full	Wrist Only	Synovial Joints
Pre-Gd	nRMSE	26.30 ± 9.16	17.82 ± 6.31	260.24 ± 158.56
	PSNR	17.77 ± 0.95	22.99 ± 0.91	8.94 ± 1.64
	SSIM	0.60 ± 0.03	0.94 ± 0.00	
PatchGAN Registered	nRMSE	6.72 ± 0.81 ***	6.07 ± 1.22 ***	23.14 ± 7.37 **
	PSNR	20.77 ± 0.65 ***	25.40 ± 1.24 **	12.10 ± 1.34 **
	SSIM	0.58 ± 0.02	0.94 ± 0.01	
PatchGAN Unregistered	nRMSE	8.46 ± 1.03 ***	7.68 ± 1.41 **	28.96 ± 10.57 **
	PSNR	19.85 ± 0.69 ***	24.38 ± 1.21 *	11.23 ± 1.52 *
	SSIM	0.56 ± 0.02 *	0.94 ± 0.01	
UNet Registered	nRMSE	6.29 ± 0.88 ***	4.36 ± 0.60 ***	26.18 ± 7.45 **
	PSNR	22.03 ± 0.60 ***	27.13 ± 0.69 ***	11.58 ± 0.93 **
	SSIM	0.69 ± 0.02 ***	0.95 ± 0.00 **	
UNet Unregistered	nRMSE	7.73 ± 1.03 ***	5.38 ± 0.73 ***	29.69 ± 7.60 **
	PSNR	21.20 ± 0.62 ***	26.20 ± 0.77 ***	10.98 ± 0.87 *
	SSIM	0.68 ± 0.02 ***	0.95 ± 0.01 *	

## Data Availability

The data and codes used to generate results are available from the corresponding author upon reasonable request. Due to patient privacy concerns, the data is not publicly available.

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
