# Peer review of "Synthetic Inflammation Imaging with PatchGAN Deep Learning Networks"

_bioengineering, 2023, doi:10.3390/bioengineering10050516_

Round 1
Reviewer 1 Report
The following observations may help to improve your work in better way. These are,
1. basically this topic covers the medical image and processing techniques. The objectives of the research and existing challenges are not listed in abstract.
2. How the UNet and PatchGAN were trained?. The details are not furnished.
3. Introduction section should cover from basis to advances findings. How the new readers will understand the presented work?. Currently it covers advanced findings gathered from recent published work.
4. Page number 5, equations are not cited and epoch ranges should mention for evaluation.
5. Figure 2: Network Performance with and without Deconvolutions in Decoding Path of Generator- This figure and its data are not readable.
6. Test bed details are most important for validating results. Use the detailed specification in results section.
7. Data sets and sources are not available. It is difficult to check the obtained results.
8. Conclusion and future direction section are missing. If you use this section, it will be good.
Reviewer 2 Report
The objective of this paper is to propose a method for developing deep learning (DL) pipelines that can generate synthetic post-contrast wrist magnetic resonance (MR) images from pre-contrast images, while also utilizing image quality metrics to evaluate the diagnostic and perceptual quality of the generated images in comparison to true post-contrast images. This research has significant implications for the field of medical imaging. However, there are several areas that require modification and improvement:
- The background and purpose of the study need to be elaborated upon to provide more context for readers to understand the research. It would be helpful to provide more detailed information about the motivation for the study and the significance of the proposed approach in the broader context of medical imaging.
- The image and table captions should be clear, concise, and relevant to the content of the article. For example, the labels in Figures 2 and 3 should be modified to accurately describe the content of the images.
- The discussion section requires more depth. The authors should provide a more comprehensive discussion of the significance of their research findings, including potential implications for clinical practice, and provide suggestions for future research directions.
- The language used in the paper needs to be improved. The authors should ensure that grammar and spelling errors are corrected and that the writing adheres to academic writing guidelines.
- The authors should conduct a more thorough literature review to ensure that their work is situated within the broader context of related research. For example, the authors could consider including the following related works in their literature review: SA-GAN: Stain Acclimation Generative Adversarial Network for Histopathology Image Analysis, Direct delineation of myocardial infarction without contrast agents using a joint motion feature learning architecture, Synthetic MRI in differentiating benign from metastatic retropharyngeal lymph node: combination with diffusion-weighted imaging, DualMMP-GAN: Dual-scale multi-modality perceptual generative adversarial network for medical image segmentation, and Magnetic resonance imaging contrast enhancement synthesis using cascade networks with local supervision.
Reviewer 3 Report
The paper is written in an organized manner, however, there are major issues need to be fixed:
1. The Abstract needs to be poorly written. The background, methods, results, and conclusions should be included in the abstracts.
2. It is not clear what the challenge here and how the proposed method work well.
3. The problem statement and objective of the work should be clearly mentioned in the ‘Introduction’ section.
4. The introduction section should include a list of the major study contributions.
5. Justification need to be added for using SSIM, nRMSE, and PSNR as the main evaluation methods
6. Needs to add more details about UNet baseline and PatchGAN
7. Future work should be added to the conclusion section.
Round 2
Reviewer 2 Report
The authors have addressed all the issues raised, and I recommend an accept.
Reviewer 3 Report
The authors have satisfactorily addressed all comments